# The *Lotus angustissumus* Group (Fabaceae): Can Phylogenetic Patterns Be Accommodated by a Taxonomic Concept?

**DOI:** 10.3390/plants13010101

**Published:** 2023-12-28

**Authors:** Tatiana E. Kramina, Tamerlan R. Hadziev, Tahir H. Samigullin

**Affiliations:** 1Department of Higher Plants, Biological Faculty, Lomonosov Moscow State University, GSP-1, Leninskie Gory, 119234 Moscow, Russia; thadziev@list.ru; 2A.N. Belozersky Institute of Physico-Chemical Biology, Lomonosov Moscow State University, GSP-1, Leninskie Gory, 119991 Moscow, Russia; samigul@belozersky.msu.ru

**Keywords:** nrITS, *psb*A-*trn*H, *rps*16, *trn*L-F, phylogeny, Mediterranean, taxonomy

## Abstract

The *Lotus angustissimus* group represents an example of obvious contradictions between the molecular and morphological data. This group includes from six to eight mostly annual species of *Lotus* section *Lotus*, with the center of species diversity in the Mediterranean. We performed a phylogenetic analysis of the genus *Lotus* with an expanded representation of all known species of the *L. angustissimus* group using both nuclear (nrITS) and a set of plastid DNA markers and compared the results with traditional taxonomy of this group. Our results demonstrated a non-monophyletic nature of the study group. In addition, the nrITS and plastid dataset tree topologies were incongruent with each other in both Bayesian and ML analyses. We revealed very complex phylogenetic relationships among the species of the group. Important results obtained in this study are: (1) genetic and geographical differentiation in the latitudinal direction (between *L. angustissimus* and *L*. *praetermissus*) and in the longitudinal direction among closely related accessions identified as *L. castellanus*, *L. lourdes-santiagoi* and *L. palustris*; (2) close genetic links between the Anatolian endemic *L. macrotrichus* and *L. praetermissus* from Eastern Europe and Central Asia; and (3) the hybrid nature of *L. subbiflorus* with *L. parviflorus* as its presumed male parent species. These results are discussed in the context of morphology, biogeography and taxonomy.

## 1. Introduction

Molecular phylogenetic studies have radically changed the traditional taxonomy of many groups of flowering plants, including the Fabaceae family (e.g., [1]). In some cases, taxa isolated by classical taxonomy methods have been confirmed using molecular phylogenetic approaches. In other cases, there were obvious contradictions between the molecular and morphological data, and one such example is the *Lotus angustissimus* group. This is a relatively small group of mostly annual *Lotus* species of the *Lotus* section, which includes, according to various points of view, from six to eight species and has a center of species diversity in the Mediterranean.

### 1.1. Taxonomic Composition of the Lotus angustissimus Group

The species *Lotus angustissimus* L. was described by Linnaeus [2]. Heyn included it and a number of morphologically similar species in the *Lotus angustissimus* group [3]. She showed the importance of keel beak and fruit length features for the delimitation of species in this group. Heyn [3] separated species of the group into two subgroups: with a straight keel beak (*L. angustissimus*, *L. macrotrichus* Boiss., *L. suaveolens* Pers. and *L. parviflorus* Desf.) and a curved keel beak (*L. subbiflorus* Lag. and *L. palustris* Willd.).

Kuprianova [4] described a species *Lotus praetermissus* Kuprian. distributed in Eastern Europe and Central Asia, which differs from *L. angustissimus* in solitary erect stems with spreading branches, wider and shorter legumes, and dark brown seeds. Greuter et al. [5], in the checklist of Mediterranean flora, distinguished the following species: *L. angustissimus* (incl. *L. praetermissus*), *L. divaricatus* Boiss. (syn. *L. macrotrichus*), *L. parviflorus*, and *L. palustris* aggr., which includes *L. palustris*, *L. castellanus* Boiss. & Reut. and *L. hispidus* Desf. ex DC. (syn. *L. suaveolens*, *L. subbiflorus*).

Kramina conducted a taxonomic revision of the *Lotus angustissimus* group [6]. She demonstrated that the priority name for the species with a long straight keel beak and middle-sized pods is *L. subbiflorus* [7], not *L. suaveolens* [8]. She also argued why the name *L. hispidus* Desf. cannot be used for this taxon and demonstrated that the name *L. castellanus* Boiss. & Reut. has a priority for the species with a long, curved keel beak. She discovered a new significant character, i.e., ovary and pod pilosity along the ventral suture, which distinguishes *L. castellanus* from *L. palustris*. In *L. castellanus*, fruits and ovaries are hairy along the ventral suture and in *L. palustris*, they are glabrous or sometimes with a few trichomes [6]. Multivariate morphological study of the *L. angustissimus* group from the same work [6] demonstrated the following results: (1) *L. praetermissus* is not distinguished from *L. angustissimus* by main diagnostic features (except those of seeds); (2) *L. macrotrichus* is close to *L. angustissimus* and *L. praetermissus*; (3) there are transitional morphological forms between *L. castellanus* and *L. palustris*; (4) analysis carried out on a large set of morphological features led to the division of the dataset into four groups—a. *L. parviflorus*, b. *L. subbiflorus*, c. *L. angustissimus* + *L. praetermissus* + *L. macrotrichus*, and d. *L. castellanus* + *L. palustris*—but for a reduced set of characteristics, including only diagnostic ones, and connections were found between *L. subbiflorus* and groups a and d.

Pina and Valdés described a new species in the *L. angustissimus* group, *Lotus lourdes-santiagoi* Pina & Valdés, close to *L. castellanus* but differing by hairy standard petals. They also clarified a set of diagnostic characteristics for distinguishing species in the group [9].

In the present paper we consider the *Lotus angustissimus* group according to Pina and Valdés [9], including seven species: *L. angustissimus* L. (incl. *L. praetermissus* Kuprian.), *L. macrotrichus* Boiss., *L. subbiflorus* Lag., *L. parviflorus* Desf., *L. castellanus* Boiss. & Reut., *L. lourdes-santiagoi* Pina & Valdés, and *L. palustris* Willd. Important morphological characteristics are presented in Figure 1. Main diagnostic characteristics of species of the *Lotus angustissimus* group [3,4,6,9] are summarized in Table 1.

### 1.2. Chromosome Numbers in the Lotus angustissimus Group

Traditionally, the *Lotus angustissimus* group was included in *Lotus* section *Lotus*, which is characterized by main chromosome number x = 6, in contrast to the other sections of the genus with main chromosome number x = 7 [10]. According to literature data, species of *L. angustissimus* are diploids with 2n = 2x = 12 (*L. castellanus*, *L. parviflorus*, *L. praetermissus*), tetraploids with 2n = 4x = 24 (*L. subbiflorus*) or have two ploidy levels 2n = 12, 24 (*L. angustissimus*) or variable chromosome numbers 2n = 12, 14, 24 (*L. macrotrichus*, *L. palustris*) [3,10,11,12,13]. The number 2n = 14 indicated for the species of *L. angustissimus* group is doubtful. In general, the chromosome numbers published for species of the *L. angustissimus* group should be treated with caution due to the difficulties in determining the species in this group and the different taxonomic views of the authors.

### 1.3. Geographical Distribution of Lotus angustissimus Group

Data on the geographical distribution of the species of the *Lotus angustissimus* group are summarized from several sources [3,6,14]. *Lotus angustissimus* is distributed in the Mediterranean, Europe (except the northern part), and the Middle East. The distribution area of *Lotus praetermissus* is in Eastern Europe and Central Asia (Kazakhstan, South Ural and South Siberia). *Lotus macrotrichus* is endemic to Western Anatolia and adjacent islands. Of a pair of closely related species, *Lotus castellanus* is common in the Western Mediterranean, and *Lotus palustris* is sporadically found throughout the Mediterranean and in Western Europe, although the exact range of the latter is unknown due to poorly defined morphological boundaries between the two species. *Lotus lourdes-santiagoi* is endemic to Andalusia (southwestern Spain). The main distribution area of *Lotus parviflorus* and *Lotus subbiflorus* is the Western Mediterranean and Macaronesia. *Lotus subbiflorus* also was rarely recorded from the Eastern Mediterranean and southwest England.

Species of the *Lotus angustissimus* group (especially, *L. angustissimus* and *L. subbiflorus*) have spread beyond their main range and formed secondary ranges in Australia, New Zealand, South and North America, South Africa and the Pacific Islands [6,14].

### 1.4. Previous Phylogenetic Studies of Lotus angustissimus Group

Early molecular phylogenetic study of the tribe Loteae investigated phylogenetic relationships among several species of the *L. angustissimus* group by nrITS markers [15]. In this work, the group was not monophyletic: *L. angustissimus* was clustered with *L. parviflorus* in a common clade, and *L. palustris* with *L. corniculatus* in another unrelated clade. *L. subbiflorus* took a separate position. The results obtained by Allan et al. [15] were not confirmed by subsequent studies.

Degtjareva et al. [16] analyzed nrITS markers in a representative sample of *Lotus* species and revealed the separation of the section *Lotus* into two highly supported clades. The first clade included *L. subbiflorus*, *L. parviflorus* and *L. conimbricensis*, and the second all the other studied species of the section, including *L. praetermissus* and *L. angustissimus*, as well as representatives of the groups *L. pedunculatus* and *L. corniculatus*. Interestingly, these two clades were not sister to each other on the phylogenetic tree [16]. Subsequent phylogenetic studies of the genus *Lotus* revealed that *L. conimbricensis* is not related to any species of the *Lotus angustissimus* group [17,18].

Phylogenetic study of a representative set of *Lotus* species based on nuclear (nrITS, nrETS) and plastid (*psb*A-*trn*H, *rps*16) DNA markers revealed subdivision of the genus *Lotus* into the so-called Southern clade and one or more (according to different markers) Northern clades [17]. The Southern clade includes the majority of sections of the genus, and the Northern one only three sections (i.e., *Lotus*, *Dorycnium* and *Bonjeanea*). The study demonstrated a non-monophyletic pattern of the section *Lotus* [17]. The species of the *Lotus angustissimus* group belong to the Northern clade (by plastid markers) or are distributed among the northern clades (by nuclear markers). None of the markers confirmed the monophyly of the *Lotus angustissimus* group as understood by Heyn [3]. Close relationships were demonstrated between *L. subbiflorus* and *L. parviflorus* and between *L. castellanus* and *L. palustris* by ITS marker, and between *L. subbiflorus* and *L. castellanus* and between *L. angustissimus* and *L. praetermissus* by plastid markers. The plastid dataset also clearly showed a close relationship between *L. angustissimus* and *L. pedunculatus* groups. However, an insufficiently presented sample of species of the *Lotus angustissimus* group does not allow the authors to draw confident conclusions about the phylogenetic relationships of the species of the group [17]. The species *L. macrotrichus* and *L. lourdes-santiagoi* have not been studied by molecular methods, and their phylogenetic position was unknown until now.

The present study was aimed at conducting a phylogenetic analysis of the genus *Lotus* with an expanded representation of all currently known species of the *Lotus angustissimus* group using both nuclear (ITS1-5.8S-ITS2), and plastid DNA markers and comparison of the results obtained with the data of the traditional taxonomy of this group.

## 2. Results

### 2.1. Taxonomic Identification of Specimens Based on Morphology

The vast majority of samples were unambiguously identified as representatives of the following species based on morphological data: *L. angustissimus* (ANG1–ANG5, ANG7–ANG11, ANG13, ANG15, ZLA, 472), *L. praetermissus* (TH4-TH7, STN6, POY, 458, 473), *L. macrotrichus* (819), *L. castellanus* (CAS1–CAS2, 417, 471), *L. lourdes-santiagoi* (LS1-LS2), *L. palustris* (PAL3–PAL5, 426, 869, 875), *L. subbiflorus* (457, 470, 475, 476, SUBB1, TH2), *L. parviflorus* (PRV1–PRV5, 469). The specimen TH1 from Israel (MW 0740247) possesses no reproductive structures (flowers, fruits). It was previously identified as *Lotus palustris* Willd.; however reexamination of this specimen led to its reidentification as *Lotus rectus* L. Voucher information is presented in Table 2. Two specimens from Spain and Portugal formerly identified as *L. palustris* (PAL1 and PAL2) were intermediate between *L. castellanus* and *L. palustris* (Table 3). Three specimens (ANG12, ANG14, and ANG17) were intermediate between *L. angustissimus* and *L. subbiflorus* in their morphological characteristics and could not be unambiguously determined (Table 4).

### 2.2. Phylogenetic Analysis of nrITS Dataset

On the nrITS phylogenetic tree, the genus *Lotus* forms a well-supported clade with several subclades (Figure 2). Among them, a Southern clade, several northern clades (i.e., *Lotus* Northern clades I, II, III and IV) and a separate branch of *Lotus glinoides* are observed (Figure 2). Species of the *Lotus angustissimus* group are clustered within two highly supported clades, *Lotus angustissimus* group clade 1 and *Lotus angustissimus* group clade 2, which are not related to each other. Clade 1 is a part of the *Lotus* Northern clade I, and clade 2 corresponds to the *Lotus* Northern clade III (Figure 2).

Within clade 1, *L. praetermissus* and *L. angustissimus* form well-supported species-specific subclades. *Lotus macrotrichus* is clustered with *L. praetermissus* taking a basal position in the common *L. praetermissus* plus *L. macrotrichus* clade. *Lotus palustris*, *L. castellanus* and *L. lourdes-santiagoi* form another clade, where specimens of the latter two species are mixed in a separate smaller subclade, and specimens of *L. palustris* specimens occupy a basal position.

Clade 2 includes a mixture of studied specimens of *L. parviflorus* and *L. subbiflorus*, as well as three specimens with uncertain taxonomic identification (ANG12, ANG14, and ANG17). In the specimen ANG17, two variants of nrITS were found, differing in position 483 of the alignment (T in clone 1/C in clone 2).

Phylogenetic analysis of the nrITS dataset confirmed that the specimen TH1 belongs to the species *Lotus rectus* L.

### 2.3. Phylogenetic Analysis of the Plastid DNA Dataset

On the phylogenetic tree constructed by the plastid DNA dataset, the genus *Lotus* is also well defined and subdivided into two large highly supported clades: the Northern clade and the Southern clade (Figure 3). In addition, the Northern clade on the phylogenetic tree, constructed by plastid data, unites all members of Northern clades I, II, III and IV, identified on the tree constructed by nrITS. *Lotus glinoides* is included in the Southern clade.

Representatives of the *Lotus angustissimus* group occupy positions in three highly supported subclades within the Northern clade, that is, in Clades A, B and C. The composition of none of these three subclades is identical to that of either clade 1 or clade 2 from the nrITS tree.

Clade A on the plastid DNA phylogenetic tree corresponds to a part of clade 1 on the nrITS tree and includes three well separated species-specific groups consisting of specimens of *L. angustissimus*, *L. praetermissus* and *L. macrotrichus*, respectively.

Clade C on the plastid DNA phylogenetic tree corresponds to a part of clade 2 on the nrITS tree and includes *L. parviflorus* and three specimens of uncertain taxonomic identification (ANG12, ANG14, and ANG17).

Clade B of the plastid DNA phylogenetic tree partially corresponds to clade 1 and clade 2 of the nrITS tree and includes *L. palustris*, *L. castellanus* and *L. lourdes-santiagoi* together with *L. subbiflorus*. Within clade B, the species *L. palustris*, *L. castellanus*, *L. lourdes-santiagoi* and *L. subbiflorus* are in an unresolved position. It is worth noting that the sister position in relation to clade B is occupied by *Lotus pedunculatus* Cav., which is not a member of the *Lotus angustissimus* group, but represents a separate group of the *Lotus* section. Clades A and (B + *L. pedunculatus*) together form a larger, well-supported clade, while clade C occupies a position away from them.

Phylogenetic analysis of the plastid dataset confirmed the belonging of the TH1 sample to the species *Lotus rectus* L.

### 2.4. Geographic Patterns Observed in the Lotus angustissimus Group

Both nrITS and studied plastid markers confirm the geographical differentiation within the pair of the closely related species *Lotus angustissimus* and *L. praetermissus*. The studied specimens of *L. angustissimus* occupy the southern part of the common range of these two species, mainly around the Mediterranean and Black Seas and in the southern part of the Caspian Sea, and specimens of *L. praetermissus* are distributed in the northern part, namely in the center and southeast of the European part of Russia, Ukraine and western Kazakhstan (Figure 2, Figure 3 and Figure 4A). The only studied specimen *L. macrotrichus* from SW Turkey is clustered together with *L. praetermissus* in both phylogenetic reconstructions, though geographically it is closer to *L. angustissimus* (Figure 2, Figure 3 and Figure 4A).

Geographic pattern was also observed within the complex of *Lotus castellanus*, *L. lourdes-santiagoi* and *L. palustris* on the nrITS phylogenetic tree. All specimens of *L. palustris* from the Eastern Mediterranean are unresolved at the base of the common clade of the complex, whereas all specimens from the Western Mediterranean, including *L. castellanus* and *L. lourdes-santiagoi*, formed a highly supported subclade. Two specimens of *L. castellanus* from Spain and Portugal formerly classified within *L. palustris* (PAL1 and PAL2), but having an intermediate morphologic pattern between typical *L. palustris* and *L. castellanus*, were also members of this western subclade (Figure 2, Figure 3 and Figure 4C). In the phylogenetic tree constructed using plastid markers, no phylogenetic or geographical structure was observed in this complex, since these markers do not allow for resolving the relationships in this group of species (Figure 3).

The distribution areas of the studied specimens of *Lotus parviflorus* and *L. subbiflorus* overlap significantly (Figure 4B), which is consistent with their clustering on phylogenetic reconstruction using the nrITS dataset (Figure 2).

## 3. Discussion

Here, we present the first molecular phylogenetic study that includes all known up-to-date species of the *Lotus angustissimus* group, analyzed by both nuclear and plastid DNA markers. Despite comparatively small datasets studied for each species, we tried to cover different parts of their ranges, and the results obtained allowed us to make several important conclusions concerning phylogenetic relationships, geography and taxonomy in this group.

The phylogenetic reconstructions obtained in this study confirmed the non-monophyletic nature of the *Lotus angustissimus* group, which was previously assumed in earlier studies of the genus *Lotus* [16,17,18]. However, all representatives of this group are connected in a complex way by common molecular features. The *Lotus angustissimus* group represents a good example of the discrepancy between nuclear and plastid phylogeny, which may indicate different trends in the evolution of nuclear and plastid lines and the presence of traces of network evolution.

### 3.1. Lotus angustissimus, L. praetermissus and L. macrotrichus

The results of this study by both nuclear and plastid markers clearly support the separation of *L. praetermissus* from *L. angustissimus*. This point of view is shared by many authors of regional floras of the territories of the former USSR, Eastern Europe and Central Asia [19,20,21,22,23,24]. The opposite concept was expressed by Heyn [3], some Western European authors [5,25,26], and botanists of the Legume Phylogeny Working Group [27] who did not support the segregation of *L. praetermissus* from *L. angustissimus*. Detailed morphometric study [6] did not reveal a clear difference between the two species; however, some important characteristics (e.g., seed surface color) were not included in the analysis due to the absence of ripe seeds in the majority of studied specimens. One of the most prominent morphological features distinguishing typical *L. praetermissus* from typical *L. angustissimus* is a specific growth pattern: *L. praetermissus* is usually a small annual plant with a prominent erect main shoot branched in the upper half, whereas *L. angustissimus* is a more branched annual or perennial plant with ascending shoots. However, a number of transitional forms are observed between the two extreme morphological variants. Moreover, when the plants of *L. praetermissus* were grown from seeds in the greenhouse with sufficient water supply, they had a habitus more resembling that of *L. angustissimus* (Kramina, unpublished data).

The genetic separation of *L. praetermissus* and *L. angustissimus* is in good agreement with their geographical distribution, and the range of *L. praetermissus* is confined to more northern and eastern regions. As a whole, the results obtained in this work can serve as an argument in favor of the allocation of two taxa in the rank of independent species. This contradicts the conclusions made earlier only on morphological grounds [6]. Apparently, the problem deserves further study with the involvement of analysis of population variability at both the morphological and molecular levels.

The only studied specimen of the local endemic of Western Anatolia, *Lotus macrotrichus*, is related to *Lotus praetermissus* based on both nuclear and plastid DNA markers. These results are more or less consistent with morphometric studies [6], where a specimen of *L. macrotrichus* was morphologically close to *L. praetermissus* and Turkish specimens of *L. angustissimus* in many characteristics, but differed from them by the shape of upper leaflets, larger flowers, styles, and fruits. Sister phylogenetic relationships between *L. macrotrichus* and *L. praetermissus* do not contradict the recognition of the former as a separate species, but a more comprehensive study is needed for a balanced decision.

### 3.2. Lotus castellanus, L. lourdes-santiagoi and L. palustris

All three species share a common morphological feature, namely, a keel with a curved keel beak. Their close relationship is confirmed by the ITS nuclear marker. Our study demonstrated that *L. palustris* is paraphyletic in relation to *L. castellanus* and *L. lourdes-santiagoi* by the nrITS marker, but plastid data cannot resolve the relationships among these three species.

The molecular phylogenetic results obtained in the present study do not allow support for the isolation of *L. lourdes-santiagoi* from *L. castellanus*. These two species can be clearly distinguished from each other, but only by one morphological feature—the presence of pubescence on a standard petal. This character is very interesting because it illustrates the phenomenon of the occurrence of morphological features of a neighboring taxon in this taxon. The pubescent back surface of standard petal is known in *Lotus* section *Pedrosia* (Lowe) Christ. (in seven species, incl. *Lotus chazaliei* H. Boissieu, *L. loweanus* Webb & Berthel., and others) and section *Rhyncholotus* (Monod) D.D. Sokoloff (all members), as well as in *Cytisopsis ahmedii* (Batt. & Pit.) Lassen, *Tripodion tetraphyllum* (L.) Fourr., and *Hammatolobium lotoides* Fenzl from the tribe Loteae. Within the species *Hammatolobium lotoides*, some plants have a pubescent standard petal, whereas the others have a glabrous one [28]. A similar variability pattern was observed within four species of *Lotus* section *Pedrosia* [29]. In the *Lotus* section *Lotus*, to which representatives of the *L. angustissimus* group belong, *L. lourdes-santiagoi* is the only example of a plant with a pubescent standard petal.

The results of nrITS phylogenetic analysis indicate that the genetic data for the three species have a clear geographical structure: samples from the Western Mediterranean (typical *L. castellanus*, atypical samples from Spain and Portugal previously identified as *L. palustris*, and *L. lourdes-santiagoi*) have a number of molecular synapomorphies that distinguish them from samples distributed in the Eastern Mediterranean (typical *L. palustris*). Such a pattern can be interpreted in at least two ways: (1) all studied samples belong to the same species; and (2) samples from the Western Mediterranean belong to *L. castellanus* (or to *L. castellanus* and *L. lourdes-santiagoi*) and those from the Eastern Mediterranean should be treated as *L. palustris*. Perhaps the use of other more rapidly evolving molecular markers could be successful in this case.

The nrITS data suggest that *Lotus palustris* from the Eastern Mediterranean is closer to a hypothetical genetic variant ancestral to the pair of species *L. castellanus* and *L. palustris*, which, however, is not confirmed by the analysis of plastid data. The studied plastid data did not allow resolution of the relationships within the group of species *L. castellanus*, *L. lourdes-santiagoi*, *L. palustris* and *L. subbiflorus*, which together form clade B in the phylogenetic tree (Figure 3). One of the reasons for this phylogenetic pattern may be the slow rate of mutation of plastid DNA sequences, which, combined with incomplete lineage sorting, may lead to a lack of isolation of plastid sequences found in each of the four species. The second possible cause of such a pattern may be hybridization events that may have occurred in the evolution of this group of four species, either in recent times or in the more distant past. Two of the species forming clade B, *L. palustris* and *L. subbiflorus*, are known as possible tetraploids [3,10,11], which can be considered an argument in favor of the second opinion.

### 3.3. Lotus subbiflorus

Our study showed that *Lotus subbiflorus* is genetically close to *L. parviflorus* based on nuclear ITS data and to the complex *L. castellanus*–*L. lourdes-santiagoi*–*L. palustris* by a set of plastid markers. Such a genetic pattern may indicate the hybrid nature of *Lotus subbiflorus*. This assumption is consistent with the data on the tetraploid chromosome number of this species, 2n = 24 [3,10], and the cytogenetic data obtained by Ferreira and Pedrosa-Harand [11]. Ferreira and Pedrosa-Harand [11], using the FISH method, demonstrated the presence of Ljcen1 signals in only one set of *L. subbiflorus* chromosomes, which allowed them to assume an allotetraploid origin of this species. According to the results of our study, the diploid *L. parviflorus* may be the presumed male parent species of *L. subbiflorus*, and a representative of *L. castellanus*–*L. lourdes-santiagoi*–*L. palustris* complex may be its putative female parent species. Data on morphology and geography tentatively suggest the participation of *L. castellanus* as a second parent species of *L. subbiflorus*. *L. castellanus* is similar to *L. subbiflorus* in long keel beak and middle-sized fruit (ca. 1.5–3 times as long as calyx), but they differ from each other in the keel beak shape, which is straight in *L. subbiflorus* and curved in *L. castellanus*. The curve of the keel tip is sometimes not very noticeable, so from time to time the two species are confused. Sympatric geographical distribution of *L. castellanus* and *L. subbiflorus* in the Western Mediterranean does not contradict the assumption of the participation of *L. castellanus* as a second parent species of *L. subbiflorus* either.

The phylogenetic position of three specimens, ANG12 from Portugal, ANG14 from Italy and ANG17 from Iran, raises questions. According to their morphological characteristics, these specimens occupy a more or less intermediate position between *L. angustissimus* and *L. subbiflorus*. By nrITS marker, they are clustered within a clade of *L. parviflorus* + *L. subbiflorus*, and by a plastid dataset, they are grouped with *L. parviflorus*. We checked these results by conducting two separate DNA isolations from ANG12 and ANG17 and several repetitions of each PCR reaction for all three specimens. The result was the same. As such, in this case, we have a real discrepancy between morphological identification and molecular data. The results of molecular studies imply the participation of *L. parviflorus* in the formation of the three individual plants, but we did not observe any morphological features of this species in the mentioned individuals, except for a straight keel beak, which is common for several species of the group. The first reason for such variability may be the hybrid nature of the three specimens with hidden morphological features of one parent species. One of the facts in favor of such a conclusion, which can be made for the sample ANG17, is the polymorphism of nrITS. However, this character does not allow identification of the putative parent species, as it was demonstrated for other cases within the genus *Lotus* (e.g., [30]). The second reason is incomplete lineage sorting, which is in good agreement with the geographical location of mentioned specimens in glacial refugia of the Mediterranean basin [31] and the southern Caspian region [32]. As such, the taxonomic status of the specimens ANG12, ANG14 and ANG17 remains uncertain. We prefer to identify them as *Lotus* spp. or *Lotus* hybrids. This problem needs further study using an expanded set of molecular markers, as well as various methods of analysis.

### 3.4. Lotus parviflorus

Heyn [3] considered *Lotus parviflorus* to be the least problematic species within the *Lotus angustissimus* group due to a set of clear diagnostic features, such as a long straight keel beak and a short fruit not exceeding the calyx. The results obtained in this work confirmed the isolated position of the clade *Lotus parviflorus* + *L. subbiflorus* within the genus *Lotus* in the nrITS phylogenetic trees, which was found in previous studies on a limited dataset [17,18]. In the phylogenetic reconstructions made in this study using the plastid DNA dataset, *L. parviflorus* forms a separate clade, without *L. subbiflorus*. The inclusion of three specimens with uncertain taxonomic identification in this clade was discussed above. Thus, there is clear evidence of a genetic relationship between *L. parviflorus* and only one representative of the *Lotus angustissimus* group, *L. subbiflorus*, which is consistent with morphological data [6]. We are not inclined to combine these two species into one, since the hybrid nature of *L. subbiflorus* is shown in the present and previous studies.

## 4. Materials and Methods

### 4.1. Plant Material

The molecular study involved 88 specimens, including 53 specimens of *Lotus angustissimus* group, 32 specimens of other *Lotus* species representing all main sections of the genus, and 3 specimens of genera *Cytisopsis*, *Hammatolobium* and *Tripodion*, closely related to *Lotus*. Samples for molecular studies were taken from herbarium specimens stored in herbaria ANK, B, BM, GAZI, LE, MA, MHA, MW, and NSW. Voucher information and GenBank accession numbers are presented in Table 2 and Appendix B. Geographical distribution of specimens included in molecular analyses is presented on a map (Figure 4) prepared using SimpleMappr online software (https://www.simplemappr.net, accessed on 2 December 2023) [33]. All the studied specimens of the *Lotus angustissimus* group were taxonomically identified by morphological features using descriptions and identification keys from the main literary sources [3,6,9].

### 4.2. DNA Extraction, Amplification and Sequencing

DNA was extracted from herbarium specimens (ca. 20 mg of leaf tissue) with a NucleoSpin Plant II kit (Macherey-Nagel, Düren, Germany) according to the manufacturer’s instructions or using the CTAB method [34]. The nrDNA ITS and plastid DNA regions *psb*A-*trn*H intergenic spacer, *trn*L-*trn*F intergenic spacer (IGS) and *trn*L intron, and *rps*16 intron were selected for the analysis because of their utility in *Lotus* and high variability [17,35,36]. The sequences of the nrITS were amplified with primers NNC-18S10, C26A [37], ITS2 and ITS3 [38]. The amplification of the *psb*A-*trn*H spacer was conducted using primers trnH2 [39] and psbAF [40]. The sequences of the *trn*L-*trn*F region of plastid DNA were amplified using standard primers “c”, “d”, “e” and “f” [41], and the sequences of *rps*16 intron using primers rpsF, rpsR2 [42], Lot-rps16-F and Lot-rps16-intR [35]. PCRs were performed in a 0.02 mL mixture containing 10–20 ng DNA, 3.2 pmol of each primer and MasDDTaqMIX (Dialat LTD, Moscow, Russia) containing 0.2 mM of each dNTP, 1.5 mM MgCl_2_, and 1.5 units of SmarTaqDNA polymerase. Amplification of the nrITS region and all plastid DNA regions was performed under the following conditions: hold 94 °C, 3 min; 94 °C, 30 s; 57 °C, 40 s; 72 °C, 60 s; repeat 30 cycles; extend 72 °C, 3 min. Amplification of the same regions in the samples taken from old herbarium specimens was performed under the following program: hold 94 °C, 1 min; 94 °C, 30 s; 57 °C, 40 s; 60 °C, 1 min 20 s; repeat 35 cycles; 57 °C, 40 s; 60 °C, 1 min 20 s; repeat 2 cycles.

PCR products were purified using a Cleanup Mini kit (Evrogen, Moscow, Russia) and then used as a template in sequencing reactions with the ABI Prism BigDye Terminator Cycle Sequencing Ready Reaction Kit v. 3.1. Sequencing was performed on the 3730 DNA Analyzer (Life Technologies, Carlsbad, CA, USA) in the Syntol company (Moscow, Russia). Forward and reverse strands of all samples were sequenced. For the majority of samples, the polymorphism of ITS within one specimen was detected by direct sequencing (without cloning), by the presence of double peaks on electropherogram. ITS sequences of the specimen ANG17 were cloned in the *E. coli* vector and then sequenced in the Evrogen Joint Stock Company (Moscow, Russia).

The sequences were aligned using MAFFT version 7.215 [43,44] and then adjusted manually in BioEdit version 7.2.5 [45]. The matrices of *psb*A-*trn*H spacer, *trn*L-F and *rps*16 intron plastid DNA regions were concatenated into a single matrix. Gap-rich and ambiguous positions were excluded from the analyses. The aligned data matrices are presented in Appendix A.

### 4.3. Phylogenetic Analyses

Phylogenetic analyses were performed separately for the nrITS dataset and a concatenated plastid DNA dataset using both maximum likelihood and Bayesian inference methods.

Maximum likelihood analyses were performed in IQ-tree version 1.6.12 [46], and internal branch support was assessed using the ultrafast bootstrap [47] with 10,000 resamplings. The GTR + F + G4 model of nucleotide substitutions for plastid data and the SYM + G4 model for nrITS were selected as the most appropriate based on the Bayesian information criterion in the built-in ModelFinder utility [48].

Bayesian inference was performed using MrBayes v. 3.2.6 [49] considering the optimal model of nucleotide substitutions selected by AICc in PAUP version 4.0a [50] for each marker: SYM + Γ (symmetrical model with substitution rate heterogeneity) for nrITS, and GTR + Γ for plastid data. The Bayesian analysis used four independent runs of 25 million generations and four chains sampling every 1000th generation. Non-convergence assessment and burn-in estimation was carried out in VMCMC ver. 1.0.1 [51]. The first two million generations were discarded as burn-in, and the remaining trees from both runs were combined in a 50% majority-rule consensus tree.

## 5. Conclusions

The results of the present phylogenetic study of the *Lotus angustissimus* group together with a representative set of *Lotus* species, using nrITS and a set of three plastid DNA regions, clearly demonstrated a non-monophyletic nature of the study group. The nrITS and plastid dataset tree topologies were incongruent in both Bayesian and ML analyses.

Our data demonstrated geographical differentiation between genetic variants within a pair of species *Lotus angustissimus* L. and *L. praetermissus* Kuprian., and the southern variants corresponded to the first and the northern ones to the second of the species. The phylogenetic position of the Anatolian endemic *Lotus macrotrichus* Boiss. has been determined for the first time, which turned out to be close to *L. praetermissus*, but separated from the latter by a number of mutations both in nrITS and plastid sequences. The position of *Lotus subbiflorus* Lag. in phylogenetic trees confirms its hybrid nature and implies that its presumed male parent is *L. parviflorus* Desf. The present phylogenetic results do not allow us to support the isolation of *L. lourdes-santiagoi* from *L. castellanus*, which is consistent with the very weak morphological difference between these taxa. The nrITS data indicate a clear geographical structure within the *L. castellanus*–*L. lourdes-santiagoi*–*L. palustris* complex, where the samples of the Western Mediterranean differ from those of the Eastern Mediterranean.

The present study has revealed several trends in genetic variability within the *Lotus angustissimus* group. In order to assess the limits of this variability and the degree of genetic isolation of individual species, it is necessary to expand the study both with respect to the sample (including the population level) and with respect to the studied genetic markers.

## Figures and Tables

**Figure 1 plants-13-00101-f001:**
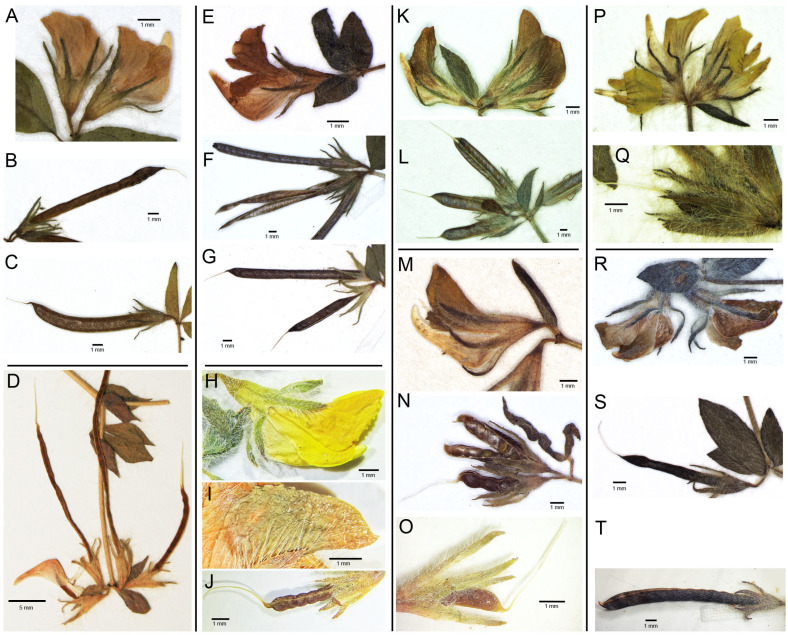
Morphological characteristics of species of the *Lotus angustissimus* group. (**A**–**C**) *Lotus praetermissus* (herbarium specimen MG1: European Russia, Volgograg region, T. Kramina, MW), (**A**) umbel at anthesis, (**B**,**C**) ripe fruits; (**D**) *Lotus macrotrichus* (herbarium specimen 819: Turkey, H. and E. Walter 375, B), unripe fruits and keel petal; (**E**–**G**) *Lotus angustissimus*, (herbarium specimen: Azerbaijan, Lerik, V. Egorov, MHA 0246632), (**E**) flower, (**F**) ripe fruit, (**G**) unripe fruit; (**H**–**J**) *Lotus lourdes-santiagoi* (herbarium specimen LS1: Spain, Cádiz, S. Silvestre et B. Valdés 2548, MA), (**H**) flower, (**I**) pubescence on standard petal, (**J**) unripe fruit pubescent along the ventral suture; (**K**,**L**) *Lotus subbiflorus* (herbarium specimen 470: Italy, M. Iberite 15222, MHA), (**K**) umbel at anthesis, (**L**) fruits; (**M**–**O**) *Lotus castellanus* (herbarium specimens: Spain, prov. Ávila, Mombeltrán, A. Segura-Zubizarreta 38.109 and 38.111 (sample 471), MHA), (**M**) flower, (**N**) ripe fruit, (**O**) unripe fruit pubescent along the ventral suture; (**P**,**Q**) *Lotus parviflorus* (herbarium specimen 469: Spain, Toledo, A. Segura Zubizarreta 34.567, MHA), (**P**) umbel at anthesis, (**Q**) unripe fruit; (**R**–**T**) *Lotus palustris* (herbarium specimens: Israel, Philistaean Plain, 14.7.1926, M. Zohary, and 22.06.1958, M. Zohary and I. Amdursky, MHA), (**R**) umbel at anthesis, (**S**) unripe fruit, (**T**) ripe fruit.

**Figure 2 plants-13-00101-f002:**
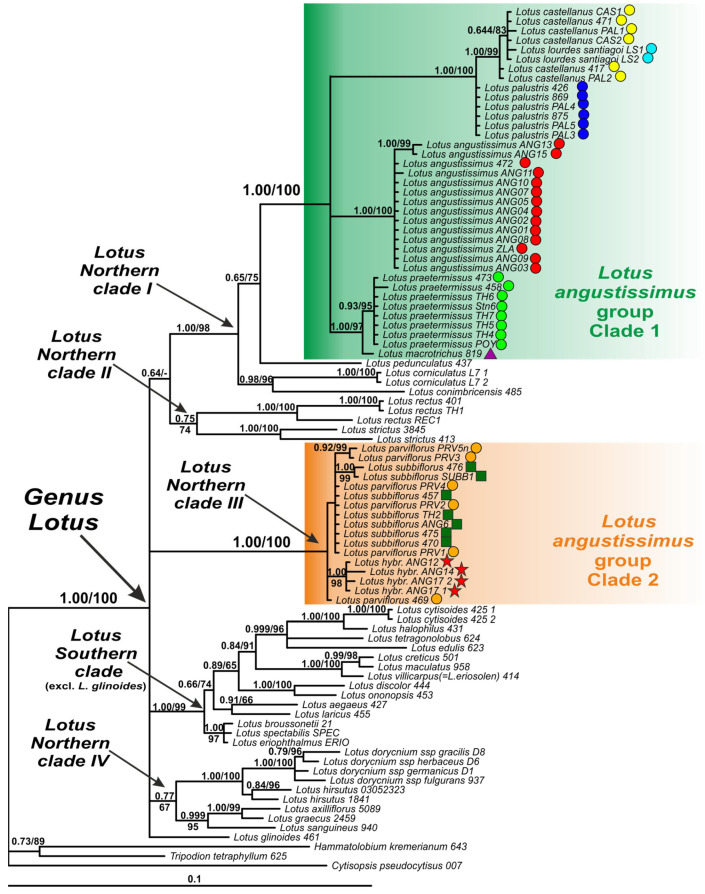
Phylogenetic relationships in *Lotus* with expanded representation of the *L. angustissimus* group inferred from Bayesian analysis of the nrITS dataset. Branch lengths are proportional to the number of expected nucleotide substitutions: scale bar corresponds to 0.1 substitutions per site. Numbers above branches are posterior probabilities. Numbers below branches or after slashes are bootstrap support values found in maximum likelihood (ML) analysis of the same dataset (values equal or more than 0.6/60% shown). The colored symbols after the names of the samples indicate belonging to different species or hybrids. See Table 2 and Appendix B for voucher information.

**Figure 3 plants-13-00101-f003:**
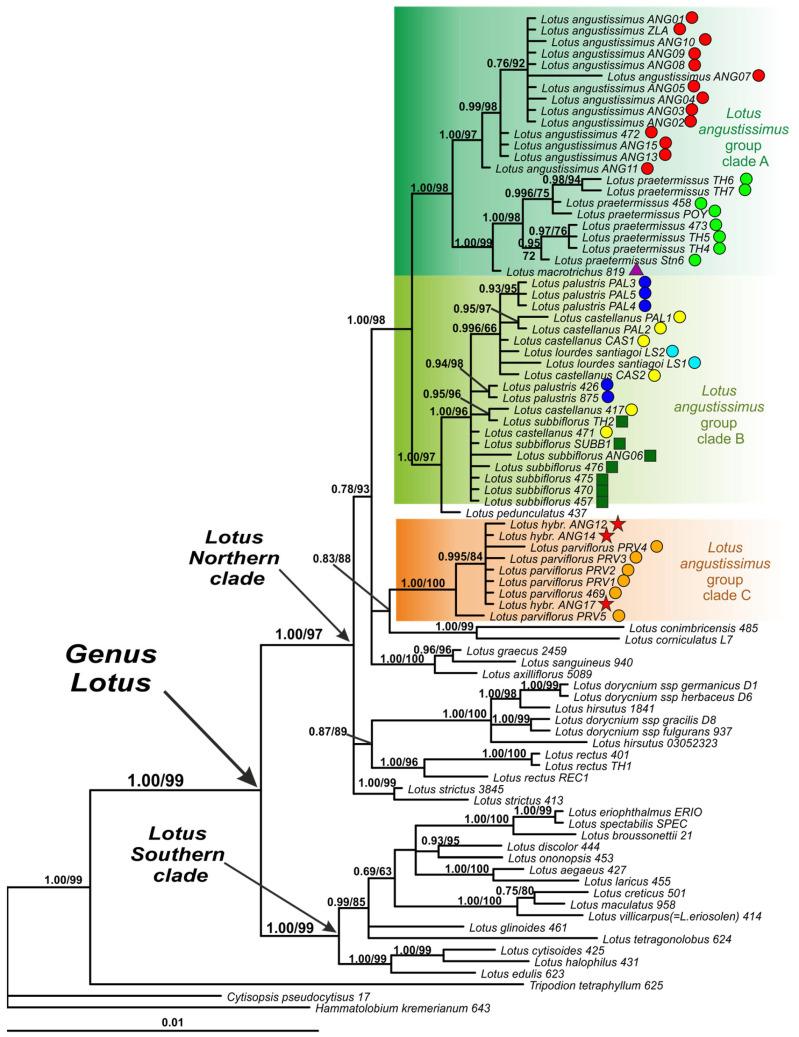
Phylogenetic relationships in *Lotus* with expanded representation of the *L. angustissimus* group inferred from Bayesian analysis of the plastid DNA dataset. Branch lengths are proportional to the number of expected nucleotide substitution: scale bar corresponds to 0.01 substitutions per site. Numbers above branches are posterior probabilities. Numbers below branches or after slashes are bootstrap support values found in ML analysis of the same dataset (values equal or more than 0.6/60% shown). The colored symbols after the names of the samples indicate belonging to different species or hybrids. See Table 2 and Appendix B for voucher information.

**Figure 4 plants-13-00101-f004:**
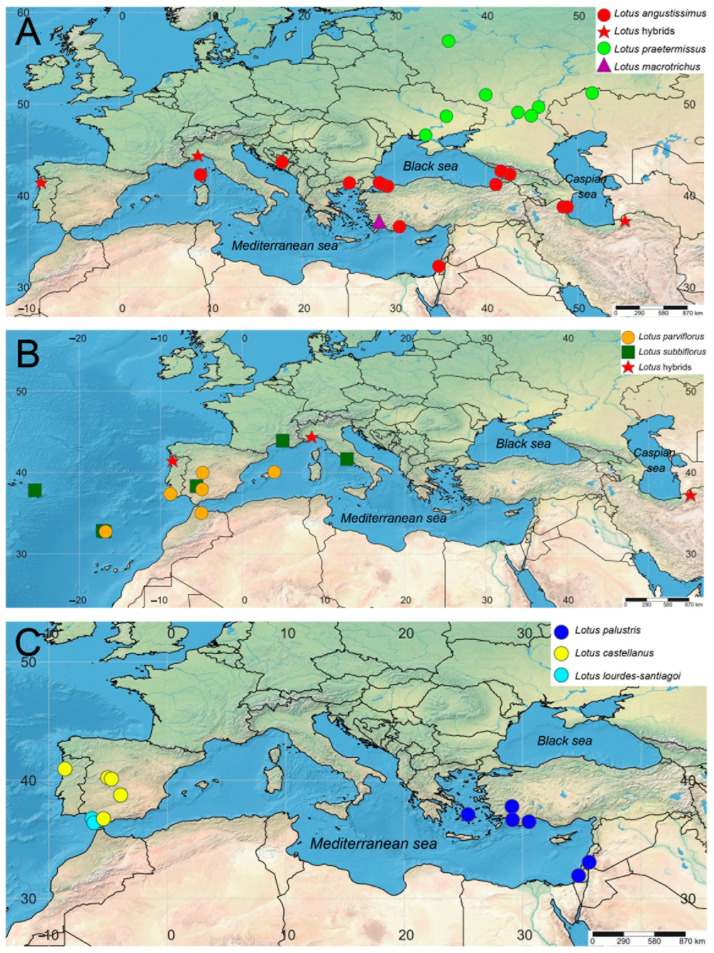
Geographical localities of specimens of the *Lotus angustissimus* group studied here using molecular methods: (**A**) *L. angustissimus*, *L. praetermissus*, *L. macrotrichus*, and *Lotus* hybrids, (**B**) *L. parviflorus*, *L. subbiflorus*, and *Lotus* hybrids, (**C**) *L. palustris*, *L. castellanus*, and *L. lourdes-santiagoi*. See Table 2 and Appendix B for voucher information.

**Table 1 plants-13-00101-t001:** Morphological characteristics of species of the *Lotus angustissimus* group.

Species	*L. angustissimus*	*L. praetermissus*	*L. macrotrichus*	*L. subbiflorus*	*L. parviflorus*	*L. castellanus*	*L. lourdes-santiagoi*	*L. palustris*
Keel beak	Straight, short		Straight, middle-sized	Straight, long		Curved		
Fruit length	Long ((2) 3–9 times as long as calyx)			Middle-sized (1.5–3.5 times as long as calyx)	Short (± as long as calyx)	Middle-sized (1.2–3.3 times as long as calyx)		Middle-sized or long (1.6–4.5 times as long as calyx)
Ovary pilosity	glabrous					Hairy at least in the upper half		Hairy in the upper third or fourth or glabrous
Style length	2.3–3.3 mm	2.5–3 mm	5.5–6.5 mm	2.9–4.4 mm	3.2–3.9 mm	3.8–4.9 mm	3.8–4.9 mm	3.2–5.1 mm
Standard back	glabrous						hairy	glabrous
Density of indumentum	low	medium or high	high	medium	high	low	medium	high
Life form	annual or perennial	annual	annual	annual, biennial or perennial	annual	annual or perennial	annual	annual
Chromosome numbers (2n)	12, 24	12	12, 14 (?), 24	24	12	12	?	12, 14 (?), 24

**Table 2 plants-13-00101-t002:** Taxa, sample code, voucher information and GenBank accession numbers of *Lotus angustissimus* group specimens used in molecular analyses. Herbarium codes according to Index Herbariorum. New sequences indicated by an asterisk. Accessions numbers without an asterisk were taken from GenBank.

Sample Code: Voucher Information (Herbarium Code); Coordinates	ITS	*trn*l-F	*rps*16	*psb*A-*trn*H
*Lotus angustissimus* L.				
472: Australia, Norfolk Island, Collins Head Road, 14.10.1999, B.M. Waterhouse BMW 5510 (NSW); 29.042 S, 167.969 E	DQ166243	MF158217	KT262868	KT262798
ANG01: Turkey, C3 Antalya Kemer, Faselis koyu, 23.06.1978, H. Pesmen 4035 et al. (GAZI); 36.602 N, 30.5601 E	OR789491 *	OR825410 *	OR825408 *	OR920533 *
ANG02: Turkey, A2(A) Istanbul, Pasaköy, Omerli Baraj, 30.06.2001, E. Yurdakulol 3729 (ANK); 41.0163 N, 29.2758 E	OR789492 *	OR825411 *	OR825409 *	OR920534 *
ANG03: Turkey, A8 Rize: Ardeşen-Findikli, 23.05.1981, A. Güner 3591 (ANK); 41.2201 N, 41.0686 E	OR794120 *	OR825412 *	OR842897 *	OR920535 *
ANG04: Turkey, Istanbul, 2 km W of Istanbul Airport, 26.05.2019, T. Kramina, M. Lysova s.n. (MW); 41.2444 N, 28.6667 E	OR794121 *	OR825413 *	OR842898 *	OR920536 *
ANG05: Turkey, Istanbul, near Karacaköy, 26.05.2019, T. Kramina, M. Lysova s.n. (MW); 41.4147 N, 28.3722 E	OR804011 *	OR863133 *	OR863092 *	OR920537 *
ANG07: Georgia, Tsqaltubo near Kutaisi, 28.10.1935, V. Komarov s.n. (LE); 42.3271 N, 42.6003 E	OR804012 *	OR863134 *	OR863093 *	OR920538 *
ANG08: Azerbaijan, N of Lerik, 25.07.1963, A.V. Bobrov, N.N. Tzvelev 1010 (LE); 38.7734 N, 48.4153 E	OR804013 *	OR863135 *	OR863094 *	OR920539 *
ANG09: Abkhasia, Gal’sky distr., near Achigvara, 23.06.1989, A. Dolmatova, V. Dorofeev 2656 (LE); 42.683 N, 41.6366 E	OR804014 *	OR863136 *	OR863095 *	OR920539 *
ANG10: Azerbaijan, S of the mouth of Lenkoranchay river, 10.07.1963, A.E. Bobrov, N.N. Tzvelev 27 (LE); 38.7505 N, 48.8692 E	OR804015 *	OR863137 *	OR863096 *	OR920541 *
ANG11: Israel, Sharon Plain, Netanya, 29.04.1964, C. Heyn s.n. (MHA); 32.3085 N, 34.8595 E	OR804016 *	OR863138 *	OR863097 *	OR920542 *
ANG13: France, Corsica, Forêt d’Aïtone bei Evisa, 25.07.1932, Paul Aellen 175 (LE); 42.2765 N, 8.8491 E	OR804017 *	OR863139 *	OR863098 *	OR920543 *
ANG15: Hercegovina, supra vicum Jablanica, 07.1933, Sillinger et Deyl s.n. (LE); 43.6605 N, 17.7573 E	OR804018 *	OR863140 *	OR863099 *	OR920544 *
ZLA: Bulgaria, M. Rhodope Orientalis, prope urbem Zlatograd, 23.07.1954, N. Stojanov et al. s.n. (MW); 41.383 N, 25.1 E	OR804019 *	MF158219	OR863100 *	OR920545 *
*Lotus praetermissus* Kuprian.				
458: Russia, Volgograd region, Bykovo distr., 1 km to N of Krasnoseletz, Moguta, 20.07.1993, T.E. Kramina s.n. (MW); 49.6726 N, 45.7392 E	DQ166227	MF158223	KT262912	KT262842
473: Ukraine, Kherson region, 15–18 km to the E from the village Novaya Mayachka, 05.06.1984, N.N. Tzvelev et al. 1630 (MW); 46.607 N, 33.4123 E	DQ168370	MF158218	OR863119 *	OR920564 *
POY: W. Kazakhstan, Uralskaya Prov., near Poyma railway station, 17.08.1952, S.A. Nikitin s.n. (MW); 51.2 N, 51.5833 E	OR804033 *	MF158220	OR863120 *	OR920565 *
Stn6: Russia, Volgograd region, Sredneakhtubinsky distr., Standartnyy village, 20.06.2008, T. Kramina Stn6 (MW); 48.7169 N, 44.9406 E	OR804034 *	MF158221	OR863121 *	OR920566 *
TH4: Russia, Volgograd region, Upper Golubaya river, 21.06.1939, P. Smirnov s.n. (MW); 49.0875 N; 43.4809 E	OR804035 *	OR863156 *	OR863122 *	OR920567 *
TH5: Russia, Tver region, Tverskoy district, 03.10.1988, A.A. Notov s.n. (MW); 56.8596 N, 35.9119 E	OR804036 *	OR863157 *	OR863123 *	OR920568 *
TH6: Russia, Voronezh region, village Varvarinki, 28.06.1948, S.V. Golitsyn s.n. (MW); 51.0046 N, 39.9776 E	OR804037 *	OR863158 *	OR863124 *	OR920569 *
TH7: Ukraine, Dnepropetrovsk region, Novomoskovsky district, Kocherezhki village, 26.06.1982, Yu. Alekseev s.n. (MW); 48.6688 N, 35.6808 E	OR804038 *	OR863159 *	OR863125 *	OR920570 *
*Lotus macrotrichus* Boiss.				
819: Turkey, Anatolien, Tahir Muğla nach Yatağan, 12.06.1955, H. und E. Walter 375 (B); 37.2250 N, 28.3416 E	OR804026 *	OR863147 *	OR863108 *	OR920553 *
*Lotus lourdes-santiagoi* Pina and Valdés				
LS1: Spain, Cádiz, Sanlúcar de Barrameda, Pinar de Algaida. 09.07.1968, S. Silvestre and B. Valdés 2548 (MA); 36.7737 N, 6.3556 W	OR804024 *	OR863145 *	OR863106 *	OR920551 *
LS2: Spain, Cádiz, Municipio de Chiclana de la Frontera, Estero La Isleta, 18.05.2013, P. Barbera and A. Quintanar 749 PB (MA); 36.4122 N, 6.1897 W	OR804025 *	OR863146 *	OR863107 *	OR920552 *
*Lotus palustris* Willd.				
426: Turkey, Antalya, near village Tekirova, 17.10.1999, S.R. Majorov s.n. (MW); 36.5010 N, 30.5255 E	DQ160275	OR863148 *	OR863109 *	OR920554 *
869: Greece, Naxos, 24.10.1991, N. Böhling 582 (B); 37.1027 N, 25.3804 E	KT250896	-	-	-
875: Israel, 22.06.1958, M. Zohary and I. Amdursky s.n. (B); 31.9331 N, 34.7048 E	KT250897	OR863149 *	OR863110 *	OR920555 *
PAL3: Turkey, Denizli suburb, ruderal place, 22.07.1999, V.D. Bochkin s.n. (MHA); 37.785 N, 29.0837 E	OR804027 *	OR863150 *	OR863111 *	OR920556 *
PAL4: Turkey, C2 Muğla, Fethiye, J. Akman 15120 (ANK); 36.6606 N, 29.126 E	OR804028 *	OR863151 *	OR863112 *	OR920557 *
PAL5: Flora Palaestine, Hula Plain, W. of Gonen Marsh, 24.05.1963, M. Zohary and U. Plitman 62457 (LE); 33.0699 N, 35.5998 E	OR804029 *	OR863152 *	OR863113 *	OR920558 *
*Lotus castellanus* Boiss. and Reut.				
417: Portugal, Beira Litoral, 12.07.1977, Malato-Beliz and J.A. Guerra 13585 (MW); 40.9634 N, 8.6477 W	DQ160272	MF158216	OR863101 *	OR920546 *
471: Spain, prov. Ávila, Mombeltrán, 01.07.1990, A. Segura Zubizarreta 38.111 (MHA); 40.2169 N, 5.0298 W	DQ166238	MF158215	KT262873	KT262803
CAS1: Spain, Ciudad Real: Argamasilla de Calatrava, 26.10.2001, M. Bellet et al. RGC431 (MA); 38.7524 N, 3.9446 W	OR804020 *	OR863141 *	OR863102 *	OR920547 *
CAS2: Spain, Toledo, Hinojosa del Monte, 15.07.1977, A. Segura Zubizarreta 15.112 (LE); 40.1043 N, 4.7225 W	OR804021 *	OR863142 *	OR863103 *	OR920548 *
PAL1: Spain, Cádiz: Algodonales. Rio guadalete. Puerto de la Nava., 21.07.1983, A. Aparicio s.n. (MA); 36.7562 N, 5.3915 W	OR804022 *	OR863143 *	OR863104 *	OR920549 *
PAL2: Portugal, Beira Litoral: Barrinha de Esmoriz, 12.07.1977, Malato-Beliz and J.A. Guerra 13585 (MA); 40.9634 N, 8.6477 W	OR804023 *	OR863144 *	OR863105 *	OR920550 *
*Lotus parviflorus* Desf.				
469: Spain, Toledo, Loc. Talavera de Ca Reina: Gamonal, 09.05.1987, A. Segura Zubizarreta 34.567 (MHA); 39.9567 N, 4.8409 W	DQ166230	MF314955	MW498357	OL753560
PRV1: Morocco: c. 10 km W of Bab Berred along road to Bab Taza, 28.05.2002, S.L. Jury et al. 19385 (MA); 35.0193 N, 4.9857 W	MN545738	MN553708	OR863114 *	OR920559 *
PRV2: Portugal, Algarve, Rogil, 20.04.2011, T. Buira, J. Calvo and L. Mauro TB1529 (MA); 37.3633 N, 8.8006 W	MN545739	MN553709	OR863115 *	OR920560 *
PRV3: Spain, Islas Baleares, Menorca, 01.05.2013, J.L. Fernandez-Alonso, JFA30449 (MA); 40.0372 N, 3.9783 E	OR804030 *	OR863153 *	OR863116 *	OR920561 *
PRV4: Spain, Córdoba, La Jarosa, alt. 450 m, 03.05.1977, A. Segura Zubizarreta 14.988 (LE); 37.9067 N, 4.9103 W	OR804031 *	OR863154 *	OR863117 *	OR920562 *
PRV5: Madeira, Machico, 04.04.1985, J.R. Press s.n. (BM); 32.7314 N, 16.791 W	OR804032 *	OR863155 *	OR863118 *	OR920563 *
*Lotus subbiflorus* Lag.				
457: Australia, New South Wales, 15.03.1992, P.G. Kodela et al. 163 (NSW); 35.1537 S, 150.7002 E	DQ166237	MF158214	OR863127 *	OR920572 *
470: Italy, prov. Latina, Lazio, Pianura Pontina, 15.06.1991, M. Iberite 15222 (MHA); 41.6420 N, 12.9153 E	DQ166231	MF158212	KT262925	KT262855
475: France, dep. Vaucluse, Avignon, 20.06.1988, G. Dutartre 570 (MHA); 43.8896 N, 5.0116 E	DQ168369	OR863161 *	OR863128 *	OR920573 *
476: Spain, Soria, Granja de Torrehermosa (Badajoz), 17.05.1987, A. Segura Zubizarreta 34.566 (MHA); 38.3099 N, 5.597 W	OR913548 *	OR863162 *	OR863129 *	OR920574 *
ANG06: Madeira, Ribeira de Janela to Seixal, 17.06.1985, J.R. Press s.n. (BM); 32.8469 N, 17.156 W	OR804040 *	OR863163 *	OR863130 *	OR920575 *
SUBB1: Madeira, Levada do Furado, 23.10.1984, M.J. Short 117 (BM); 32.735 N, 16.8861 W	OR913549 *	OR863164 *	OR863131 *	OR920576 *
TH2: Azores, São Miguel Island, 04.04.2013, A. Seregin, I. Seregina s.n. (MW); 37.799 N, 25.4854 W	OR804041 *	OR863165 *	OR863132 *	OR920577 *
*Lotus* hybrids				
ANG12: Portugal, Mouquim, 08.06.1973, A. Fernandes et al. 12403 (MHA); 41.4350 N, 8.5274 W	OR913550 *	OR920527 *	OR920524 *	OR920530 *
ANG14: Italy, Liguria occid., Varazze, 18.06.1908, L.G. Gresino s.n. (LE); 44.3654 N, 8.5721 E	OR913551 *	OR920528 *	OR920525 *	OR920531 *
ANG17: Iran, Astrabad [Gorgan] prov., S of Gumbet [Gonbad-e Kavus], 01.05.1914, A. Michelson s.n. (LE); 37.2475 N, 55.1705 E	OR913552 *, OR913553 *	OR920529 *	OR920526 *	OR920532 *
*Lotus rectus* L.				
TH1: Israel, Hula Plain, 29.01.2014, A. Seregin et al. A903 (MW); 33.0688 N, 35.597 E	OR804039 *	OR863160 *	OR863126 *	OR920571 *

**Table 3 plants-13-00101-t003:** Diagnostic morphological characteristics of *Lotus castellanus*, *L. palustris* and intermediate specimens. Median and lower–upper quartile values (in brackets) are given for each quantitative character for “pure” species (taken from the previous study [6]). Observed values are given for the specimens PAL1 and PAL2. Character states for each character are indicated as follows: C—like in *L. castellanus*, P—like in *L. palustris*, C/P—intermediate, ?—unknown.

	*Lotus castellanus*	PAL1 Spain (MA)	PAL2 Portugal (MA)	*Lotus palustris*
Density of indumentum	Low	High	Medium	High
	C	P	C/P	P
Ovary pilosity along the ventral suture	In the upper 1/2	glabrous	Unknown	In the upper 1/3–1/4 or glabrous
	C	P	?	P
Style length, mm	4.45 (4.3–4.7)	3.5	4.5	4.1 (3.8–4.4)
	C	P	C	P
Fruit length, mm	9.5 (7.2–13)	24	No fruits	17 (14–18)
	C	P	?	P
Fruit-calyx index	1.98 (1.5–2.5)	2.6	No fruits	2.81 (2.6 –3.1)
	C	P	?	P
Terminal leaflet length in mm	9.55 (7.8–11.5)	7.65 (7.5; 7.8)	10.8 (9.7; 12)	14.1 (12–17.4)
	C	C	C/P	P
Taxonomic identification based on morphology	*L. castellanus*	Intermediate individual closer to *L. palustris*	Intermediate individual closer to *L. castellanus*	*L. palustris*

**Table 4 plants-13-00101-t004:** Diagnostic morphological characteristics of *Lotus angustissimus*, *L. subbiflorus* and intermediate specimens. Median and lower–upper quartile values (in brackets) are given for each quantitative character for “pure” species (taken from the previous study [6]). Observed values are given for the specimens ANG12, ANG14 and ANG17. Character states for each character are indicated as follows: A—like in *L. angustissimusus*, S—like in *L. subbiflorus*, A/S—intermediate.

	*Lotus angustissimus*	ANG14 Italy (LE)	ANG17 Iran (LE)	ANG12 Portugal (MHA)	*Lotus subbiflorus*
Keel length, mm	3.8 (3.6–4.1)	4.0	4.4	4.1	4.4 (4–4.6)
	A	A/S	S	A/S	S
Keel beak length, mm	1.3 (1.2–1.8)	1.6	2.1	2.1	2.3 (2.2–2.5)
	A	A	S	S	S
Number of flowers in umbel	1.5 (1.5–1.5)	3 (2; 4)	1.5 (1; 2)	2 (1; 3)	2.5 (2–2.5)
	A	S	A	A/S	S
Style length, mm	2.9 (2.7–3.2)	3.0 (2.8; 3.2)	3.1 (2.7; 3.5)	3.6 (3.5; 3.7)	3.9 (3.5–4)
	A	A	A	S	S
Number of seeds in pod	22 (20–26)	17	12	13; 14	11 (9 –12)
	A	A/S	S	A/S	S
Fruit length, mm	20.5 (16–22)	18 (16; 20)	16.75 (13.5; 20)	15	11 (11–14)
	A	A	A	A/S	S
Keel index	0.36 (0.34–0.43)	0.41	0.48	0.59	0.55 (0.51–0.58)
	A	A	A/S	S	S
Fruit-calyx index	3.45 (3–4.8)	3 (2.67; 3.33)	2.7	3	2.37 (2.3–2.6)
	A	A	A/S	A	S
Taxonomic identification based on morphology	*L. angustissimus*	Intermediate individual closer to *L. angustissimus*	Intermediate between *L. subbiflorus* and *L. angustissimus*	Intermediate between *L. subbiflorus* and *L. angustissimus*	*L. subbiflorus*

## Data Availability

Data are contained within the article or Appendix A.

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
