# Peer review of "The Lotus angustissumus Group (Fabaceae): Can Phylogenetic Patterns Be Accommodated by a Taxonomic Concept?"

_plants, 2023, doi:10.3390/plants13010101_

Round 1
Reviewer 1 Report
Comments and Suggestions for Authors
The manuscript is well written, original and very interesting. I added minor suggestions directly on the MS.

Author Response
We are very grateful to both reviewers for their valuable comments and advices. We tried to answer to each of the comments and also added some explanations to the text. We believe that these corrections will improve the quality of the article and make it more understandable to readers.
Answer to the Reviewer 1.
Thank you very much for your valuable comments. We have made some corrections in the text.
Corrections
Page 1, line 2: 'Leguminosae' replaced by 'Fabaceae'.
Page 1, line 14: 'Lotus angustissimus group' replaced by 'L. angustissimus group'.
Page 1, line 18: 'Genetic' replaced by 'genetic'.
Page 1, line 19: 'Lotus angustissimus' replaced by 'L. angustissimus'.
Page 1, line 19: 'L. praetermissus' in italics.
Page 1, line 21: 'Lotus macrotrichus' replaced by 'L. macrotrichus'.
Page 1, line 22: 'Lotus subbiflorus' replaced by 'L. subbiflorus'.
Page 1, line 25: Key words: 'Lotus angustissimus; Lotus castellanus; Lotus palustris; Lotus subbiflorus;' deleted; 'taxonomy' added.
Page 1, line 29: 'Leguminosae' replaced by 'the Fabaceae family'.
Page 1, line 34: 'Lotus' in italics.
Page 1, line 37: 'combined' replaced by 'included'.
Page 1, line 44: 'Lotus praetermissus' replaced by 'L. praetermissus'.,
Page 2, line 141: author's names deleted.
Page 2, line 146: 'Lag.' and 'Pers.' deleted.
Page 2, line 148: 'Lotus lourdes-santiagoi' replaced by 'L. lourdes-santiagoi'.
Page 2, line 151: 'Lotus angustissimus' replaced by 'L. angustissimus'.
Page 2, lines 152-154: author's names deleted.
Page 4. Table 1 has been merged.
Page 17, line 492. Lotus subbiflorus - in italics.
Page 17. The reviewer's comment: In my opinion, according to molecular study, this species should be regarded as synonym of L. parviflorus.
The answer: Thank you very much to the reviewer for this comment. However we cannot agree with the reviewer's opinion on the status of Lotus subbiflorus. We keep the position to consider Lotus subbiflorus and Lotus parviflorus as two separate taxa. Our arguments in favor of such a decision are the following: 1) Lotus subbiflorus is tetraploid (and most likely allotetraploid), whereas L. parviflorus is diploid. 2) The two species are genetically close to each other by nrITS, but they are very different by a set of plastid DNA markers. This fact supports the hypothesis of the hybrid nature of Lotus subbiflorus.
Page 17. The reviewer's comment: You sampled a population in central Italy (Latium) named as L. subbiflorus. In Italy only L. parviflorus occurred (see Bartolucci et al. 2018 ) and not the putative female parent species.
The answer: Thank you very much for this comment. In 'An updated checklist of the vascular flora native to Italy' by F.Bartolucci et al. (2018), three species of the Lotus angustissimus group are included: L. angustissimus L., L. hispidus DC. and L. parviflorus Desf. We suppose, that the authors of this checklist use the name Lotus hispidus DC. instead of Lotus subbiflorus Lag. T.Kramina in her article devoted to the taxonomy of the Lotus angustissimus group (2006) argued why the name Lotus hispidus DC. or Lotus hispidus Desf. is not correct for the species with a long straight keel beak and a pod 2-3 times longer than the calyx. At this stage of the study, we consider Lotus subbiflorus to be the valid name, and Lotus hispidus to be a synonym. We realize that the relationships between the two species (i.e., L. parviflorus and L. subbiflorus) are not completely understood and resolved, and a more extensive study needs to be conducted in the future in order to draw stronger conclusions about the status of these species.

Reviewer 2 Report
Comments and Suggestions for Authors
This paper clarifies the phylogenetic relationships of the L. angustissimus group, which is difficult to classify among the Lotus sect. Lotus species, using nuclear and chloroplast genes. I think this paper contains significant results and worth publish on Plants. However, the fact that the paper cannot be understood without referring to every term used in the authors' previously published papers (e.g., Nothern Clade, Southern Clade, etc.). It should be improved.
There is no Figure 1, are two Figure 2 in this paper.
L145-149 It is not good that "Northern Clades" and "Southern Clades" are used without citation or explanation. In the next sentence, paper 17 is cited, so we assume it comes from reference paper 17, but it is not good that we do not know what paper is being discussed until then, so the paper should be specified. Also, since "Northern Clade" and "Southern Clade" will be used frequently, it would be better to have an explanation of what Northern Clade and Southern Clade are and a diagram showing where the boundary between the northern and southern clades is.
L189-190. What is "southern clade", and what is "northern lineage"?
L192 "the northern group of lineages" are the same meaning of northern lineage?
L386 "・・・・not confirmed by the analysis of plastid data."
Instead, the authors should also consider and discuss why the plastid data results did not support the hypothesis that Lotus palustris from the Eastern Mediterranean is a hypothetical ancestal to a pair of species L. castellanus and L. palustris.
L400 the author should be more specific in describing the distribution and morphology of L. castellanus and why they consider it to be the parent species of the three species.
Comments on the Quality of English Language
I didn't feel any problems with the English writing.
Author Response
We are very grateful to both reviewers for their valuable comments and advices. We tried to answer to each of the comments and also added some explanations to the text. We believe that these corrections will improve the quality of the article and make it more understandable to readers.
Answer to the Reviewer 2.
Reviewer’s comments: This paper clarifies the phylogenetic relationships of the L. angustissimus group, which is difficult to classify among the Lotus sect. Lotus species, using nuclear and chloroplast genes. I think this paper contains significant results and worth publish on Plants. However, the fact that the paper cannot be understood without referring to every term used in the authors' previously published papers (e.g., Nothern Clade, Southern Clade, etc.). It should be improved.
---- Thank you very much. We have tried to correct some parts of the text according to your recommendations. In several cases, we tried to explain our position and expanded the discussion of some ideas.
There is no Figure 1, are two Figure 2 in this paper.
--- Thank you. The figure number has been corrected.
L145-149 It is not good that "Northern Clades" and "Southern Clades" are used without citation or explanation. In the next sentence, paper 17 is cited, so we assume it comes from reference paper 17, but it is not good that we do not know what paper is being discussed until then, so the paper should be specified. Also, since "Northern Clade" and "Southern Clade" will be used frequently, it would be better to have an explanation of what Northern Clade and Southern Clade are and a diagram showing where the boundary between the northern and southern clades is.
--- Thank you very much for this comment. We added the citation [17] after the first mention of "Southern Clade" and "Northern Clade(s)". In the cited paper [17] and in the present paper, in the nrITS phylogenetic tree the clade of the genus Lotus is subdivided into one Southern clade (exluding Lotus glinoides) and several Northern clades. And on the tree constructed using a set of plastid markers, the Lotus clade is subdivided into one Southern clade and one Northern clade.
L189-190. What is "southern clade", and what is "northern lineage"?
---Thank you for this comment. We have marked the four Northern clades on the Figure 2. We have also added some additional explanations to the text.
L192 "the northern group of lineages" are the same meaning of northern lineage?
--- The text has been corrected.
L386 "・・・・not confirmed by the analysis of plastid data."
Instead, the authors should also consider and discuss why the plastid data results did not support the hypothesis that Lotus palustris from the Eastern Mediterranean is a hypothetical ancestal to a pair of species L. castellanus and L. palustris.
--- Thank you very much for this comment. We added an additional text with the explanation why the plastid data do not allow to resolve the relationships in the group of species L. castellanus, L. palustris, L. lourdes-santiagoi and L. subbiflorus.
L400 the author should be more specific in describing the distribution and morphology of L. castellanus and why they consider it to be the parent species of the three species.
--- Thank you very much for this comment. We have expanded the discussion of this idea.
Comments on the Quality of English Language. I didn't feel any problems with the English writing.
